# Rhotekin regulates axon regeneration through the talin–Vinculin–Vinexin axis in *Caenorhabditis elegans*

**Yoshiki Sakai, Tatsuhiro Shimizu, Mayuka Tsunekawa, Naoki Hisamoto\*, Kunihiro Matsumoto**\*

Division of Biological Science, Graduate School of Science, Nagoya University, Chikusa-ku, Nagoya, Japan

\* g44177a@nucc.cc.nagoya-u.ac.jp (KM); i45556a@cc.nagoya-u.ac.jp (NH)

**Data Availability Statement:** The authors confirm that all data underlying the findings are fully available without restriction. All relevant data are

## Abstract

Axon regeneration requires actomyosin interaction, which generates contractile force and pulls the regenerating axon forward. In *Caenorhabditis elegans*, TLN-1/talin promotes axon regeneration through multiple down-stream events. One is the activation of the PAT-3/integrin–RHO-1/RhoA GTPase–LET-502/ROCK (Rho-associated coiled-coil kinase)–regulatory non-muscle myosin light-chain (MLC) phosphorylation signaling pathway, which is dependent on the MLC scaffolding protein ALP-1/ALP-Enigma. The other is mediated by the F-actin-binding protein DEB-1/vinculin and is independent of the MLC phosphorylation pathway. In this study, we identified the *svh-7*/*rtkn-1* gene, encoding a homolog of the RhoA-binding protein Rhotekin, as a regulator of axon regeneration in motor neurons. However, we found that RTKN-1 does not function in the RhoA–ROCK–MLC phosphorylation pathway in the regulation of axon regeneration. We show that RTKN-1 interacts with ALP-1 and the vinculin-binding protein SORB-1/vinexin, and that SORB-1 acts with DEB-1 to promote axon regeneration. Thus, RTKN-1 links the DEB-1–SORB-1 complex to ALP-1 and physically connects phosphorylated MLC on ALP-1 to the actin cytoskeleton. These results suggest that TLN-1 signaling pathways coordinate MLC phosphorylation and recruitment of phosphorylated MLC to the actin cytoskeleton during axon regeneration.

## Author summary

Axon regeneration requires actomyosin interactions, whereby actomyosin generates contractile forces to advance the regenerating axon. In *C. elegans*, TLN-1/talin promotes axon regeneration through the PAT-3/integrin–RHO-1/RhoA–LET-502/ROCK–MLC phosphorylation signaling pathway, which is dependent on the MLC phosphorylation scaffold protein ALP-1/ALP-Enigma. TLN-1 also activates the F-actin-binding protein DEB-1/vinculin in axon regeneration, but this pathway activates SORB-1/vinexin but does not induce MLC phosphorylation. Here, we identified the *svh-7*/*rtkn-1* gene, which encodes a homolog of the RhoA-binding protein Rhotekin, as a regulator of axon regeneration. However, we show that RTKN-1 does not function in the RhoA–ROCK–MLC phosphorylation pathway in the regulation of axon regeneration. Instead, RTKN-1 can connect phosphorylated MLC

within the paper and its Supporting Information files.

**Funding:** This work was supported by grants from the Ministry of Education, Culture and Science of Japan (19H00979 to KM and 21H02578 to NH). YS and TS were supported by a research fellowship from the Japan Society for the Promotion of Science Research Fellowship. The funders had no role in study design, data collection and analysis, decision to publish, or preparation of the manuscript.

**Competing interests:** The authors have declared that no competing interests exist.

on ALP-1 to the actin cytoskeleton by physically linking the DEB-1–SORB-1 complex to ALP-1. This study suggests that the TLN-1–DEB-1–SORB-1–RTKN-1 axis promotes axon regeneration by recruiting myosin to the actin cytoskeleton via ALP-1.

## Introduction

Axon regeneration after neuronal injury is a biologically conserved process that restores functions to the nervous system. Injured axons form new growth cones that pull and extend the axon along the extracellular matrix (ECM). The driving force behind this axonal growth is a non-muscle myosin-dependent contractile force generated by actomyosin interactions [1]. This contractile force is transmitted to the ECM via cell–ECM adhesion sites, allowing axons to grow along the ECM [2]. Two molecular events take place during this process. One is the phosphorylation of the non-muscle myosin light-chain (MLC), which activates the myosin ATPase activity required for force generation [3]. This phosphorylation is mediated by several kinases, including MLC kinase (MLCK), Rho-associated coiled-coil kinase (ROCK) and citron kinase [4–6]. Another is the recruitment of myosin to the adhesion-associated actin cytoskeleton to transmit the forces generated to the ECM. It is unclear how these events are coordinated in time and space to achieve effective axon regeneration.

The cytoskeletal protein talin plays a central role in force generation by linking the actin cytoskeleton to the ECM [7]. Talin activates integrins and vinculin, proteins essential for cell–ECM adhesion. Integrins are receptors that bind to the ECM and normally exist in an inactive conformation with low binding affinity to the ECM [8–10]. Upon activation by talin, integrins undergo a conformational change from an inactive to an active state and bind tightly to the ECM, where they transduce intracellular signals via protein kinases and GTPases to regulate myosin activity [11]. Vinculin is a multi-domain cytoskeletal adaptor protein that, upon binding to talin, interacts and cross-links with F-actin, which then interacts with the talin–integrin–ECM complex to form an adhesion-bound actin cytoskeleton [12,13]. Activated vinculin also recruits other cytoskeletal proteins such as vinexin and paxillin to regulate adhesion strength and dynamics [14–17].

The nematode *Caenorhabditis elegans* has recently emerged as an attractive model for studying the mechanisms of axon regeneration in the mature nervous system [18,19]. Its amenability to genetic analysis has led to the discovery and elucidation of novel signaling pathways involved in axon regeneration. We have previously shown that the RHO-1/RhoA GTPase–LET-502/ROCK–MLC-4/MLC phosphorylation signaling pathway in *C. elegans* promotes axon regeneration in a manner dependent on the ALP (α-actinin-associated LIM protein)-Enigma protein ALP-1 (Fig 1A; [20]). Genetic analysis revealed that ALP-1 serves as a scaffold for activation of MLC phosphorylation via the RHO-1–LET-502 pathway. In addition, axon injury induces inside-out activation of integrins via the cAMP–EPAC-1/Epac (Exchange protein directly activated by cAMP)–RAP-2/Rap GTPase–TLN-1/talin cascade, leading to activation of the RHO-1–LET-502 pathway (Fig 1A; [21]). Thus, TLN-1 ultimately induces MLC-4 phosphorylation through integrin activation. In addition to its role as an integrin activator, TLN-1 also activates DEB-1/vinculin, an essential linker protein between the actin cytoskeleton and ECM-bound integrins [22,23]. Indeed, DEB-1 is required for axon regeneration in *C. elegans* [21]. However, expression of MLC-4(DD), which mimics phosphorylation, does not suppress the *deb-1* defect [21], suggesting that the TLN-1–DEB-1 signaling pathway regulates axon regeneration independently of the MLC-4 phosphorylation pathway (Fig 1A).

We have previously genetically identified a number of genes (*svh*) that regulate axon regeneration [24]. In this study, we analyzed the *svh-7/rtkn-1* gene, which encodes a homolog of

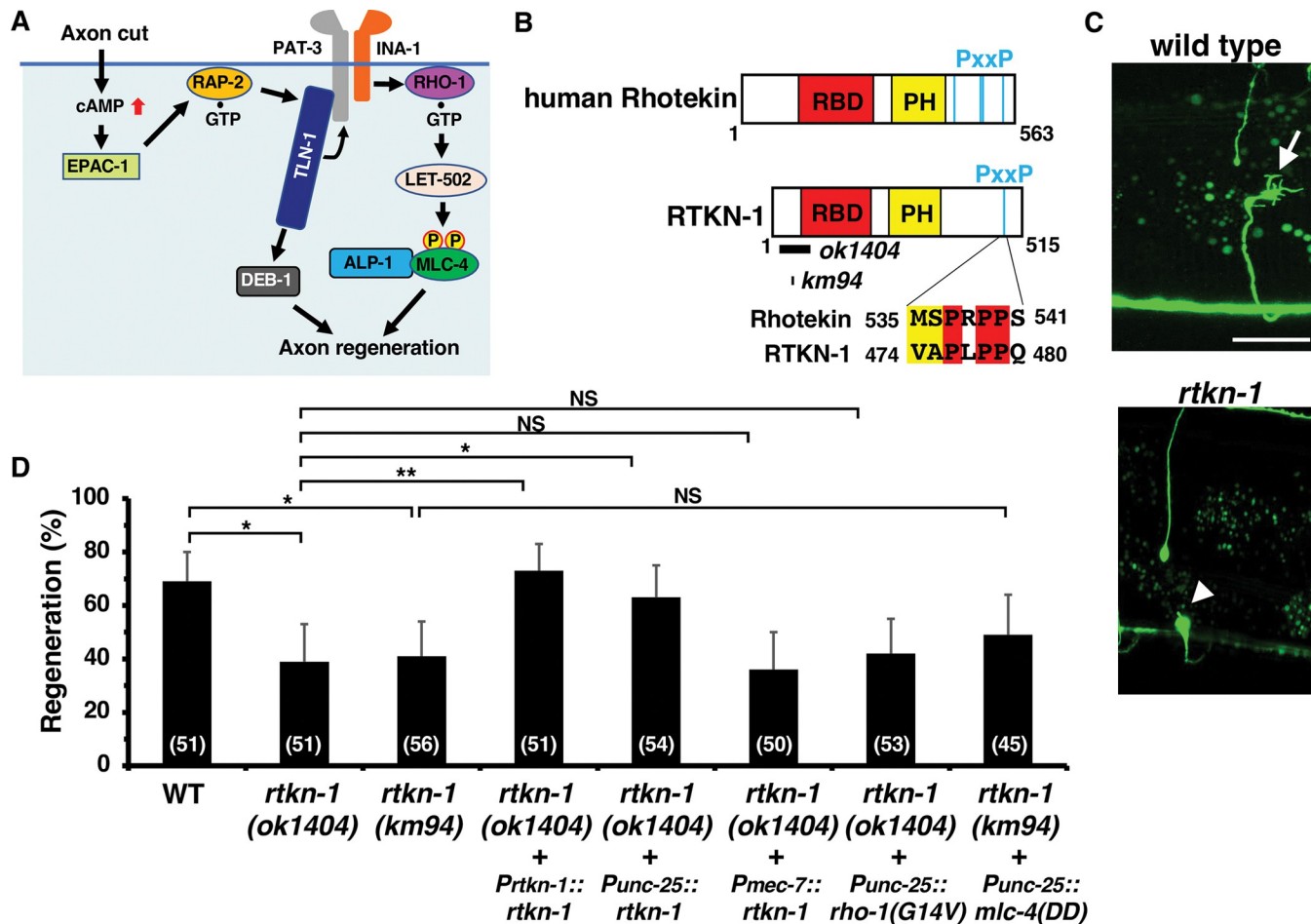

**Fig 1. SVH-7/RTKN-1 is required for axon regeneration.** (A) Regulation of axon regeneration by the TLN-1/talin signaling network. In response to axon injury, cAMP levels are elevated, resulting in the activation of EPAC-1, which then activates RAP-2. GTP-bound RAP-2 interacts with and activates TLN-1, which then activates PAT-3 and DEB-1. Activation of PAT-3 leads to phosphorylation of MLC-4 via the RHO-1–LET-502 pathway. ALP-1 acts as a platform for LET-502-mediated phosphorylation of MLC-4. DEB-1 acts independently of the MLC-4 phosphorylation pathway. (B) Domain structure of RTKN-1 and human Rhotekin. The Rho binding domain (RBD) and pleckstrin homology (PH) domains are shown in red and yellow, respectively. The Pro-rich motif (PxxP) is also shown. Identical and similar residues are highlighted in red and yellow shading, respectively. The bold lines below the RTKN-1 diagram indicate the extent of the deleted regions in the *ok1404* and *km94* mutants. (C) Representative D-type motor neurons in wild-type and *rtkn-1(ok1404)* mutant animals 24 h after laser surgery. Yellow and white arrowheads indicate regenerating and non-regenerating growth cones, respectively. Scale bar, 10 μm. (D) Percentage of axons that initiated regeneration 24 h after laser surgery at the young adult stage. The number of axons examined is shown. The error bar represents the 95% confidence interval (CI). *$P < 0.05$, **$P < 0.01$, according to Fisher's exact test. NS, not significant.

mammalian Rhotekin [25]. Mammalian Rhotekin was originally identified as a putative target of RhoA [26], but we found that RTKN-1 does not function in the RhoA–ROCK–MLC phosphorylation pathway in regulating axon regeneration. Instead, we showed that RTKN-1 binds to the DEB-1 binding protein SORB-1/vinculin and ALP-1, physically linking the DEB-1–SORB-1 complex to ALP-1. These results suggest that the TLN-1–DEB-1–SORB-1–RTKN-1 axis promotes axon regeneration by recruiting phosphorylated MLC on ALP-1 to the actin cytoskeleton.

## Results

### SVH-7/RTKN-1 is homologous with mammalian Rhotekin

We have previously isolated a number of candidate genes involved in axon regeneration and named them the *svh* genes [24]. Among these, the *svh-7* gene encodes a protein homologous to

human Rhotekin (Fig 1B). The *svh-7* gene was found to be identical to the *rtkn-1* gene [25] and so *svh-7* was renamed *rtkn-1*. Rhotekin was originally identified as a putative target of RhoA [26]. Rhotekin contains an N-terminal Rho-binding domain (RBD), a central pleckstrin homology (PH) domain and a C-terminal Proline-rich motif (PxxP) (Fig 1B; [27]). The PH domain has an affinity for phosphoinositide and helps to localize proteins to specific membrane structures [28,29].

To determine whether RTKN-1 functions in axon regeneration, we examined the regeneration of laser-severed axons in γ-aminobutyric acid (GABA)-releasing D-type motor neurons (Fig 1C). In young adult wild-type animals, approximately 70% of axons initiated regeneration within 24 h of axon injury (Fig 1C and 1D, and S1 Table), whereas the frequency of axon regeneration was reduced in *rtkn-1(ok1404)* mutant animals (Fig 1C and 1D, and S1 Table). To confirm that the axon regeneration defect was due to the *rtkn-1* mutation, an approximately 14 kb DNA fragment containing the *rtkn-1* gene was introduced into animals carrying the *rtkn-1(ok1404)* mutation. We observed that the *rtkn-1* transgene, as an extrachromosomal array, significantly rescued the defect associated with this mutation (Fig 1D and S1 Table). Furthermore, we generated a new *rtkn-1* null allele, *rtkn-1(km94)* (S1 Fig), by CRISPR–Cas9 mutagenesis and confirmed the axon regeneration defect of the *rtkn-1(km94)* mutant (Fig 1D and S1 Table). These results suggest that RTKN-1 is involved in axon regeneration after laser axotomy. We next investigated whether RTKN-1 acts in a cell-autonomous manner. The *rtkn-1* defect was rescued when *rtkn-1* was expressed from the *unc-25* promoter in D motor neurons, but not when it was expressed from the *mec-7* promoter in sensory neurons (Fig 1D and S1 Table). Thus, RTKN-1 functions in a cell-autonomous manner in D-type motor neurons.

Since mammalian Rhotekin is involved in RhoA signaling [26,27,30], we examined whether RTKN-1 functions in the RhoA–ROCK–MLC phosphorylation pathway that regulates axon regeneration. We found that the expression of an active GTP-bound RHO-1, RHO-1(G14V), from the *unc-25* promoter in D-type neurons did not suppress the *rtkn-1* axon regeneration defect (Fig 1D and S1 Table). Furthermore, expression of MLC-4(DD), which mimics phosphorylation, from the *unc-25* promoter was unable to rescue the *rtkn-1* defect (Fig 1D and S1 Table). These results suggest that RTKN-1 does not function in the RHO-1–MLC-4 phosphorylation signaling pathway to regulate axon regeneration.

## RTKN-1 interacts with SORB-1/vinexin

Rhotekin has recently been shown to interact with vinexin [27]. Mammals have two major vinexin isoforms, vinexin α and vinexin β. Vinexin α is a full-length isoform with one N-terminal sorbin homology (SoHo) domain and three C-terminal Src homology 3 (SH3) domains, whereas vinexin β lacks the SoHo domain (Fig 2A; [31]). Since both isoforms interact with Rhotekin [32], the SH3 domain is responsible for the interaction with Rhotekin. The PxxP motif is an essential module mediating the interaction with the SH3 domain [33]. Consistent with this, the SH3 domain of vinexin associates with the PxxP motif and recruits Rhotekin to the midbody during cell division [32,34]. *C. elegans* SORB-1 is most homologous to vinexin (Fig 2A; [35]) and RTKN-1 has a PxxP motif in its C-terminal region (Fig 1B). We observed that SORB-1C, which contains the SH3 domain, was able to associate with RTKN-1 (Fig 2B). These results suggest that the SH3 domain of SORB-1 associates with the PxxP motif of RTKN-1, but this possibility is not yet conclusive.

## SORB-1 is involved in axon regeneration

Next, we examined whether *sorb-1* is required for axon regeneration. We observed that axon regeneration of D-type motor neurons was impaired in *sorb-1(gk304)* mutants (Figs 2C and

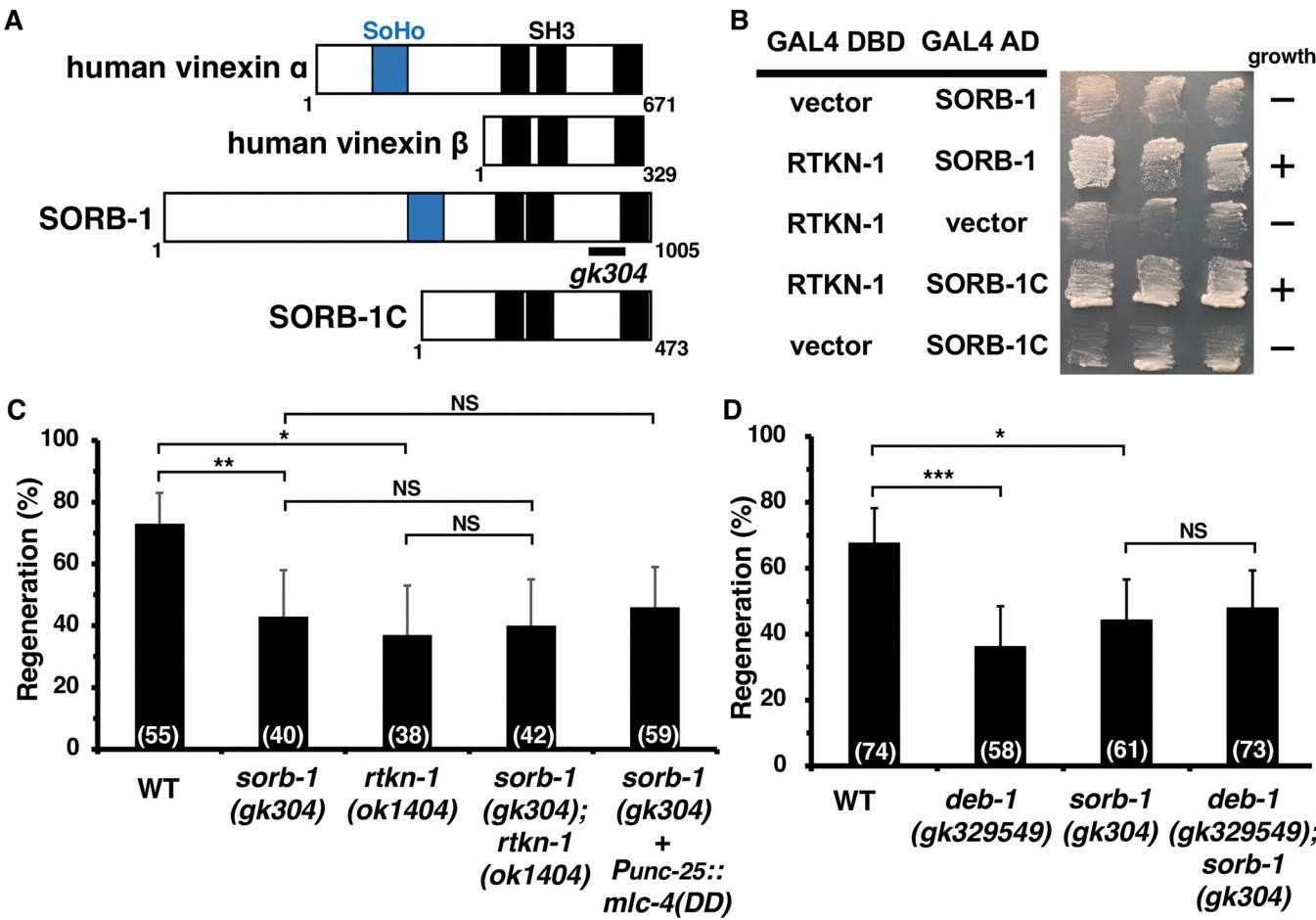

**Fig 2. SORB-1/Vinexin is required for axon regeneration.** (A) Domain structure of SORB-1 and human vinexin. The sorbin homology (SoHo) and Src homology 3 (SH3) domains are shown in blue and black, respectively. The bold line below the SORB-1 diagram indicates the extent of the deleted region in the *gk304* mutant. (B) Yeast two-hybrid assays for RTKN-1 interaction with SORB-1 and SORB-1C. The reporter strain PJ69-4A was co-transformed with the expression vectors encoding GAL4 DBD-RTKN-1, GAL4 AD-SORB-1 and GAL4 AD-SORB-1C as indicated. Yeasts carrying the indicated plasmids were grown on selective plates lacking histidine and containing 10 mM 5-aminotriazole for 4 days. (C and D) Percentage of axons that initiated regeneration 24 h after laser surgery at the young adult stage. The number of axons examined is shown. The error bar represents the 95% confidence interval (CI). *$P < 0.05$, **$P < 0.01$, ***$P < 0.001$, according to Fisher's exact test. NS, not significant.

S2, and S1 Table). We then analyzed the genetic interaction between *sorb-1* and *rtkn-1*. We found that *sorb-1(gk304)*; *rtkn-1(ok1404)* double mutants were almost as defective in axon regeneration as *rtkn-1(ok1404)* single mutants (Fig 2C and S1 Table). Furthermore, as with *rtkn-1(km94)* mutants, expression of MLC-4(DD) failed to suppress the *sorb-1(gk304)* phenotype (Fig 2C and S1 Table). These results suggest that *sorb-1* and *rtkn-1* act on the same axis to control axon regeneration in a manner independent of the MLC-4 phosphorylation pathway.

Vinexin is known as a vinculin-binding protein [31]. In mammals, activated vinculin binds to vinexin and F-actin to regulate cell adhesion and motility [13,15,17,36]. In *C. elegans*, DEB-1/vinculin also binds directly to SORB-1/vinexin [35], and *deb-1(gk329549)* mutants are impaired in axon regeneration (Fig 2D and S1 Table; [21]). SORB-1 was therefore predicted to be a potential target of the DEB-1-mediated regeneration pathway. Consistent with this, we observed that the regeneration defect in *sorb-1(gk304) deb-1(gk329549)* double mutants was not greater than the defect observed in *sorb-1(gk304)* or *deb-1(gk329549)* single mutants alone (Fig 2D and S1 Table). This suggests that DEB-1 and SORB-1 promote axon regeneration

through the same pathway. DEB-1 consists of an N-terminal head and a C-terminal tail (S3 Fig). The C-terminal tail of DEB-1, but not the N-terminal head, binds to SORB-1, consistent with mammalian vinculin and vinexin β [15]. As the *deb-1(gk329549)* allele harbors a mutation in the C-terminal tail domain where Asp-908 is replaced by valine (S3 Fig), the *deb-1 (gk329549)* mutation is predicted to be defective in its interaction with SORB-1. Taken together, these results suggest that RTKN-1 acts on the DEB-1–SORB-1 axis independently of the MLC-4 phosphorylation pathway in axon regeneration.

## RTKN-1 interacts with ALP-1

What is the role of RTKN-1 in binding to SORB-1? SORB-1 is highly localized to integrin adhesion complexes and is recruited to the actin-binding, membrane-distal regions of dense bodies in an ALP-1-dependent manner via the C-terminal SH3 domains [35]. However, SORB-1 does not directly interact with ALP-1. The SH3 domain of SORB-1 mediates the interaction with RTKN-1 (Fig 2B), raising the possibility that RTKN-1 binding to ALP-1 may link SORB-1 to ALP-1. We therefore investigated this possibility using yeast two-hybrid analysis. We found that RTKN-1 binds weakly to ALP-1 (Fig 3A). RTKN-1 adopts an autoinhibitory conformation due to intramolecular interactions between the N-terminal RBD region and the C-terminal PH domain [25]. This autoinhibitory conformation prevents RTKN-1 from interacting with its appropriate partner. Consistent with this, an RTKN-1 fragment (RTKN-1N) with an N-terminal region (amino acids 1–172) bound more strongly to ALP-1 than full-length RTKN-1 (Figs 3A and 4D). It is possible that the interaction of the C-terminal PxxP motif of RTKN-1 with the SH3 domain of SORB-1 disrupts the intramolecular self-binding of RTKN-1, resulting in the interaction of RTKN-1 with ALP-1. Taken together, these results suggest that RTKN-1 links SORB-1 and ALP-1.

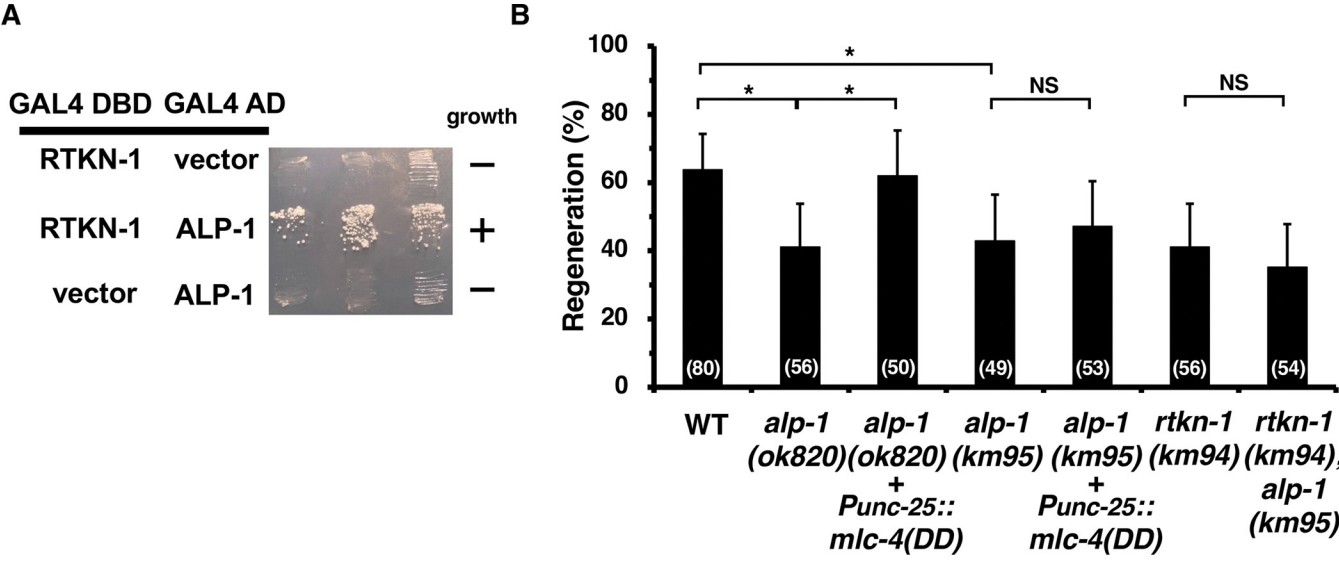

**Fig 3. Relationship between RTKN-1 and ALP-1.** (A) Yeast two-hybrid assays for RTKN-1 interaction with ALP-1. The reporter strain PJ69-4A was co-transformed with expression vectors encoding GAL4 DBD-RTKN-1 and GAL4 AD-ALP-1 as indicated. Yeasts carrying the indicated plasmids were grown on selective plates lacking histidine and containing 10 mM 5-aminotriazole for 4 days. (B) Percentage of axons that initiated regeneration 24 h after laser surgery in the young adult stage. The number of axons examined is shown. The error bar represents the 95% confidence interval (CI). *$P < 0.05$, according to Fisher's exact test. NS, not significant.

## ALP-1 is involved in both the LET-502–MLC-4 phosphorylation pathway and the SORB-1–RTKN-1 pathway

We have previously shown that ALP-1 binds to both LET-502 and MLC-4 [20], and that the axon regeneration defect in *alp-1(ok820)* mutants is rescued by expression of MLC-4(DD) (Fig 3B and S1 Table; [20]). These results suggest that ALP-1 promotes axon regeneration by acting as a scaffold for MLC-4 phosphorylation by LET-502 [20]. If ALP-1 also functions in the SORB-1–RTKN-1 signaling pathway during axon regeneration, then the axon regeneration defect caused by the *alp-1* null mutation should not be suppressed by MLC-4(DD) expression, as the *sorb-1(gk304)* and *rtkn-1(km94)* phenotypes were not rescued by MLC-4(DD) (Figs 1D and 2C). To resolve this discrepancy, we characterized the *alp-1(ok820)* allele. We found that the ALP-1 protein produced by the *alp-1(ok820)* mutant lacks the 57 amino acids preceding the LIM1 domain, but has intact LIM2–LIM4 domains (Figs 4A and S4; [37]). These results suggest that ALP-1 may function in the SORB-1–RTKN-1 pathway through the LIM2–LIM4 domains. To test this idea, we generated a null allele of *alp-1*, *alp-1(km95)* (S4 Fig). We found that axon regeneration was reduced in *alp-1(km95)* mutants to the same extent as in *alp-1(ok820)* mutants, but unlike the *alp-1(ok820)* mutant, the *alp-1(km95)* defect was not suppressed by MLC-4(DD) expression (Fig 3B and S1 Table). Furthermore, we observed that the *rtkn-1(km94); alp-1(km95)* double mutation did not enhance the axon regeneration defect of the corresponding single mutant (Fig 3B and S1 Table). Thus, ALP-1 and RTKN-1 function in the same pathway regulating axon regeneration. These results suggest that ALP-1 is involved in both the LET-502–MLC-4 phosphorylation pathway and the SORB-1–RTKN-1 pathway in the regulation of axon regeneration.

## ALP-1 interacts with LET-502, MLC-4 and RTKN-1 via the LIM2–LIM3 domains

Our genetic results raise the possibility that ALP-1 functions as a platform for both the LET-502–MLC-4 phosphorylation pathway and the SORB-1–RTKN-1 pathway. We therefore identified the sites where ALP-1 interacts with LET-502, MLC-4 and RTKN-1. Yeast two-hybrid analysis revealed that constitutively active LET-502ΔC and MLC-4 bind to the LIM2 and LIM3 domains of ALP-1 (Fig 4A–4C) and RTKN-1N binds to the LIM2 domain (Fig 4D). Thus, LET-502, MLC-4 and RTKN-1 interact with the same regions (LIM2–LIM3) within ALP-1. The *alp-1(ok820)* mutant lacks the ability to regenerate axons via the LET-502–MLC-4 phosphorylation pathway, but the SORB-1–RTKN- 1 pathway is functional. These results suggest that the partial deletion within the ALP-1 protein produced in the *alp-1(ok820)* mutant affects the MLC-4 phosphorylation process, but the exact mechanism is still unclear.

## The *tln-1(e259)* mutation affects the PAT-3 pathway but not the DEB-1 pathway

TLN-1 regulates axon regeneration through PAT-3/integrin and DEB-1/vinculin (Fig 1A; [21]). The axon regeneration defect in *pat-3(gk804163)* mutants is suppressed by expression of the phosphomimetic variant MLC-4(DD), but the *deb-1(gk329549)* deficiency is not [21]. These results are consistent with the possibility that the PAT-3 signaling pathway mediates MLC-4 phosphorylation in axon regeneration, whereas the DEB-1 signaling pathway does not (Fig 1A). However, the axon regeneration defect in *tln-1(e259)* mutants is suppressed by MLC-4(DD) expression [21], suggesting that the *tln-1(e259)* mutation affects the PAT-3 pathway but not the DEB-1 pathway. We therefore investigated which mutations occurred in the *e259* allele. DNA sequencing revealed that the *tln-1(e259)* allele carries a missense mutation

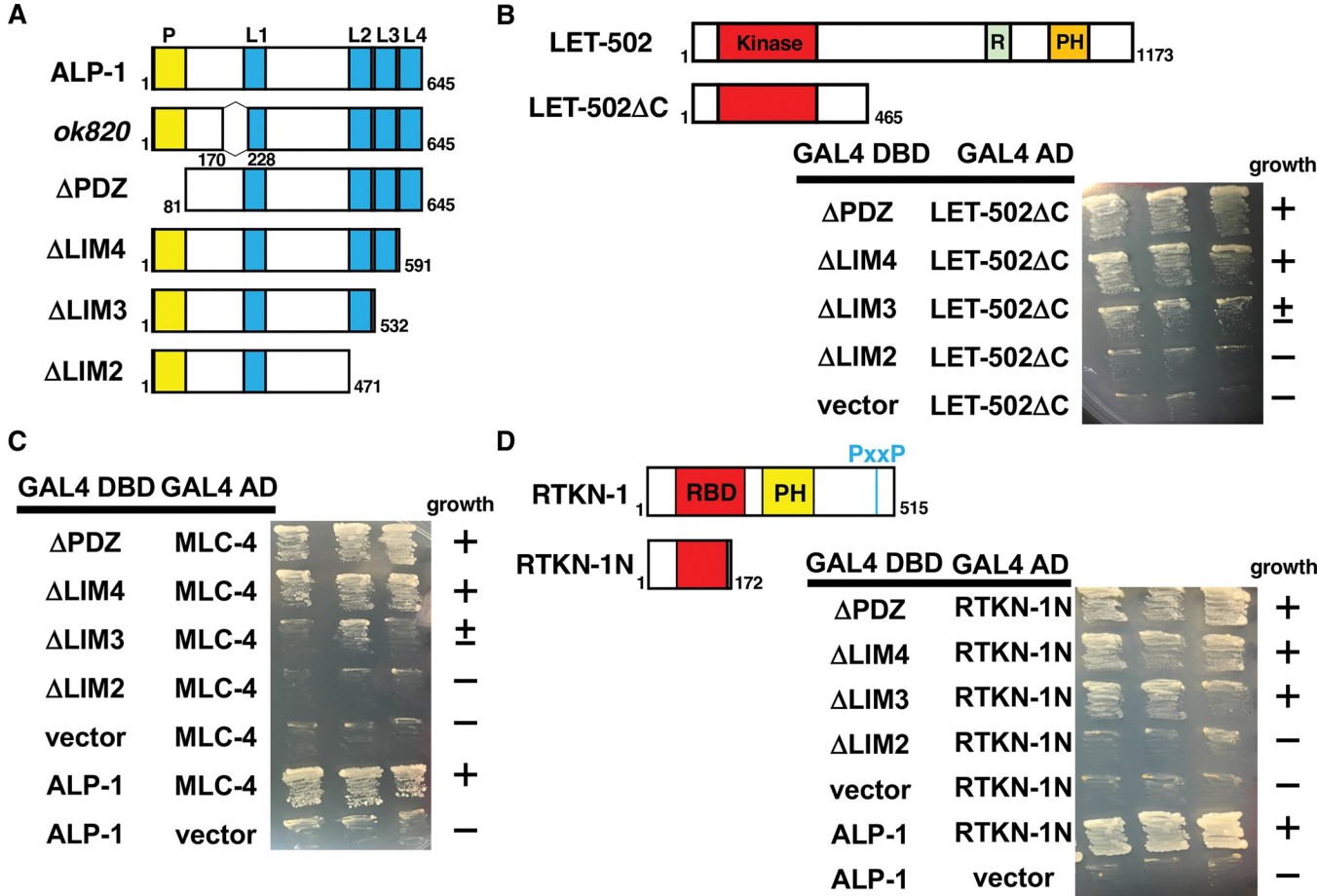

**Fig 4. Interaction of ALP-1 with LET-502, MLC-4 and RTKN-1.** (A) Structure of ALP-1. Schematic diagrams of ALP-1 are shown. The PDZ (P) and LIM domains are shown in yellow and blue, respectively. The ALP-1(Δ171–227) protein is missing 57 amino acids before the LIM1 domain. (B–D) Yeast two-hybrid assays for ALP-1 interaction with LET-502ΔC (B), MLC-4 (C) and RTKN-1N (D). Schematic diagrams of LET-502 and LET-502ΔC are shown (B). The kinase, RBD (R) and PH domains are shown in red, green and orange, respectively. Schematic diagrams of RTKN-1 and RTKN-1N are shown (D). The RBD and PH domains are shown in red and yellow, respectively. The PxxP is also shown. Yeasts carrying the indicated plasmids were grown on selective plates lacking histidine and containing 10 mM (B and D) or 5 mM (C) 5-aminotriazole for 4 days.

(*A2534T*) that replaces the conserved Ala-2534 with a threonine (Fig 5A). To confirm that the *A2534T* mutation is responsible for the reduced axon regeneration in the *tln-1(e259)* mutant, we generated a *tln-1(A2534T)* mutant at the *tln-1* locus in a wild-type background using CRISPR–Cas9 mutagenesis. We confirmed that the *tln-1(A2534T)* mutant was defective in axon regeneration (Fig 5B and S1 Table). Furthermore, in the *tln-1(e259)* mutant background, changing the Thr-2534 residue to wild-type Ala-2534 (T2534A) by CRISPR-Cas9 mutagenesis rescued the axon regeneration defect (Fig 5B and S1 Table). The *tln-1(e259)* allele was originally identified as *unc-35(e259)* [38], so *tln-1(e259)* mutants exhibit uncoordinated movement. Consistent with this, *tln-1(A2534T)* point mutants reproduced uncoordinated movement, whereas *tln-1(e259; T2534A)* mutants recovered normal movement (Fig 5C). These results support that the *A2534T* mutation is responsible for the *tln-1(e259)* phenotype.

As the *tln-1(e259)* allele has a mutation in the C-terminal dimerization domain (Fig 5A), it is possible that this mutation affects dimer formation. In fact, a recent study reported that the *L2509P* mutation (Fig 5A), located within the dimerization domain of human talin-1, disrupts dimer formation and significantly reduces adhesion maturation by integrin activation, but has

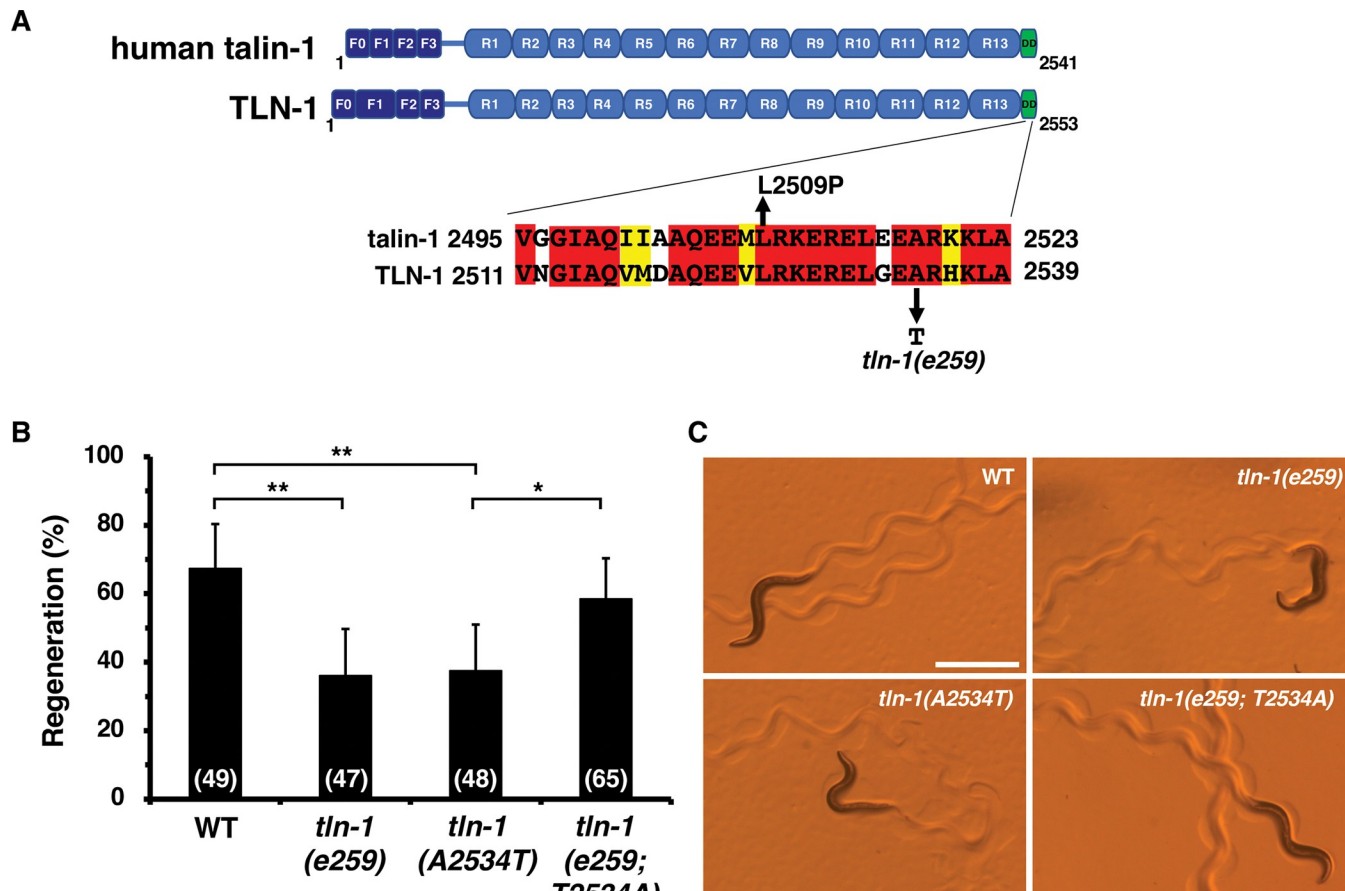

**Fig 5. The *A2534T* mutation is responsible for *tln-1(e259)* phenotypes.** (A) Domain structure of TLN-1 and human talin-1. The *tln-1(e259)* allele harbors the *A2534T* mutation in the C-terminal dimerization domain (DD). Identical and similar residues are highlighted in red and yellow shading, respectively. The *L2509P* mutation in human talin-1 is shown. The FERM (F) domain (F0–F3) and the 13 talin rod domains R1–R13 are shown. (B) Percentage of axons that initiated regeneration 24 h after laser surgery at the young adult stage. The number of axons examined is shown. The error bar represents the 95% confidence interval (CI). *$P < 0.05$, **$P < 0.01$, according to Fisher's exact test. (C) Representative trajectories of wild-type and *tln-1* mutants. Wild-type and *tln-1(e259; T2534A)* mutant animals moved in a smooth sinusoidal pattern, whereas *tln-1(e259)* and *tln-1(A2534T)* animals displayed uncoordinated movement. Scale bar, 500 μm.

little effect on its interaction with vinculin [39]. These results are consistent with the possibility that the *tln-1(e259)* mutation is only defective in the PAT-3 pathway and does not affect the DEB-1 pathway.

## Discussion

We have previously shown that TLN-1/talin promotes axon regeneration by activating the PAT-3/integrin–RHO-1/RhoA–LET-502/ROCK–MLC-4/MLC phosphorylation pathway in an ALP-1/ALP-Enigma-dependent manner [20,21]. The promotion of axon regeneration by TLN-1 also requires the DEB-1/vinculin activation pathway, which is independent of the MLC-4 phosphorylation pathway. Thus, TLN-1 successfully induces axon regeneration by simultaneously activating two different signaling pathways (Fig 6). Mammalian talin has also been shown to be involved in promoting axon growth and regeneration. Overexpression of full-length talin-1 in cultured neurons activates integrins and stimulates axon regrowth [40]. Talin consists of two domains, an N-terminal head domain that binds integrins and a C-terminal rod domain that binds vinculin [41–43]. Interestingly, the N-terminal head domain of

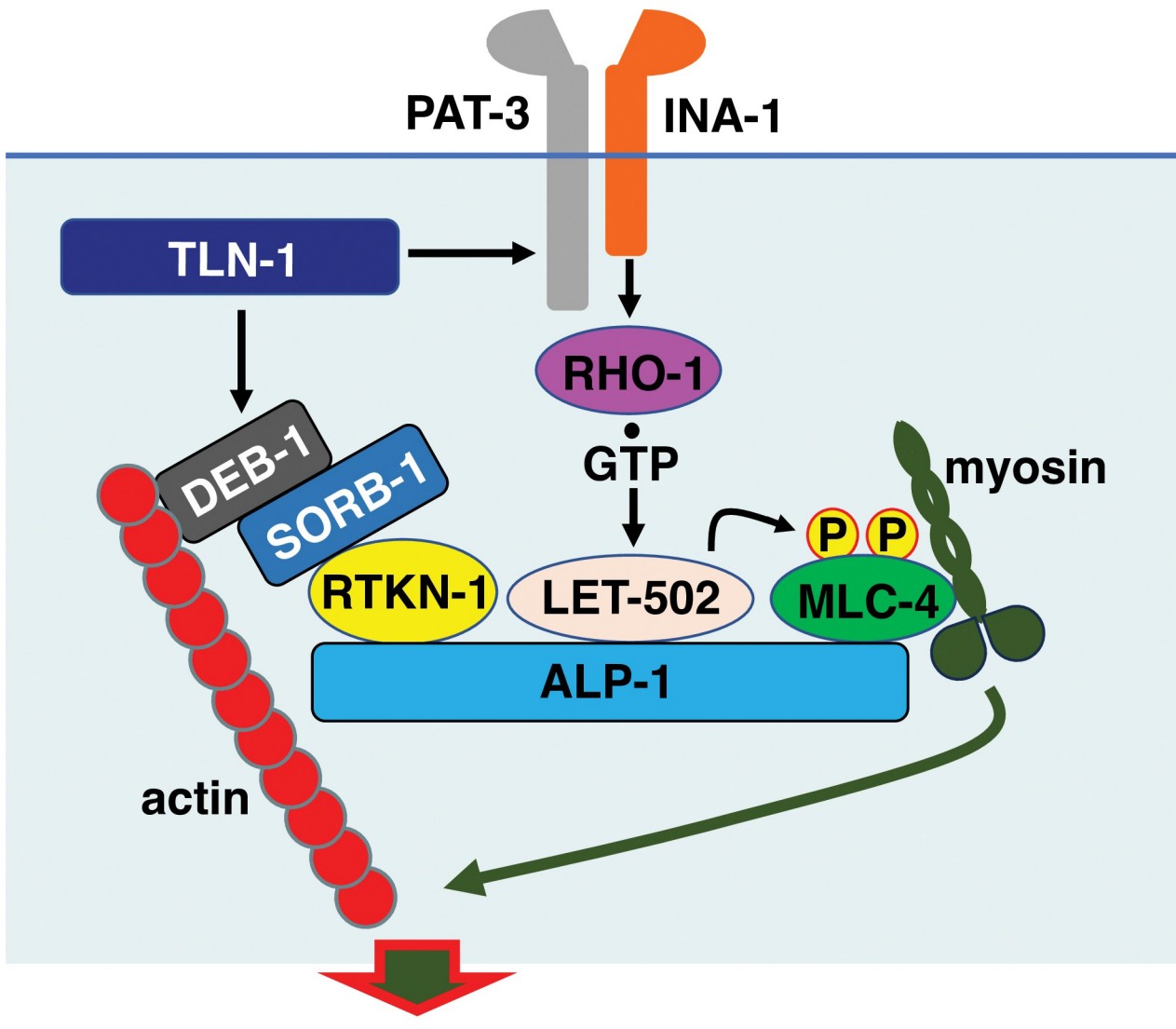

**Fig 6. Schematic model of axon regeneration regulated by the TLN-1–PAT-3 and TLN-1–DEB-1 pathways.** TLN-1 promotes axon regeneration through the PAT-3 and DEB-1 signaling pathways. The PAT-3 pathway leads to the activation of RHO-1. GTP-bound active RHO-1 activates LET-502, which binds to ALP-1 and phosphorylates MLC-4 on ALP-1. Activated DEB-1 interacts with SORB-1, leading to the binding of SORB-1 and RTKN-1. RTKN-1 then interacts with ALP-1 and recruits the DEB-1–SORB-1 complex to ALP-1. On the ALP-1 platform, phosphorylated MLC-4 binds to the adhesion-associated actin cytoskeleton and promotes actomyosin-dependent force generation that facilitates axon regeneration.

talin-1 alone is sufficient to activate integrins but not to stimulate axon regrowth, suggesting that the rod domain of talin-1 is also required for axon regeneration [40]. This result is consistent with our model that activation of both integrins and vinculin is required to promote axon regeneration. However, the function of the DEB-1/vinculin signaling pathway in regulating axon regeneration was unclear. In this study, we first attempted to identify the components that function in the TLN-1-mediated signaling pathways required for axon regeneration. As we had previously genetically identified *svh* genes that regulate axon regeneration [24], we investigated the role of the *svh-7/rtkn-1* gene, which encodes a mammalian homolog of Rhotekin [25], in regulating axon regeneration.

Mammalian Rhotekin was originally identified as a putative target of RhoA [26], but we found that RTKN-1 does not function in the RhoA–ROCK–MLC phosphorylation pathway that regulates axon regeneration. Thus, RTKN-1 acts independently of RHO-1 in axon regeneration. Mammalian Rhotekin has also been identified as a binding partner for vinexin [32], but its physiological function in this interaction remains unclear. Vinexin has been identified as a vinculin-binding protein and is required to induce a conformational change of vinculin to a form with higher affinity for actin [31,44]. The SH3 domain of vinexin binds to Rhotekin [31,32]. Similarly, RTKN-1 binds to the SH3 domain of SORB-1. RTKN-1 also binds to ALP-1, which serves as a platform for phosphorylation of MLC-4 by LET-502/ROCK [20]. Previously, we found that the *alp-1(ok820)* mutation reduced axon regeneration and that the phenotype was suppressed by phosphomimetic MLC-4(DD). These results suggest that the region deleted by the *alp-1(ok820)* mutation (57 amino acids before the LIM1 domain) is important for MLC-4 phosphorylation [20]. In contrast, the axon regeneration defect in *alp-1(km95)* null mutants is not rescued by MLC-4(DD). Thus, phosphorylated MLC-4 still requires ALP-1 to induce axon regeneration. Axon regeneration requires a myosin-dependent contractile force that is transmitted to the ECM via the cell–ECM adhesion site. This requires active myosins to interact with the adhesion-associated actin cytoskeleton, which forms the vinculin–talin–integrin–ECM complex [2]. Because RTKN-1 can bind to both SORB-1 and ALP-1, RTKN-1 can act as a bridging molecule between DEB-1–SORB-1 and ALP-1–phosphorylated MLC-4 (Fig 6). This could be a mechanism to localize phosphorylated MLC-4 to the DEB-1-associated actin cytoskeleton to induce axon regeneration. In addition, the DEB-1–SORB-1 signaling pathway and the integrin–MLC phosphorylation pathway are activated by TLN-1. Activation of these two signaling pathways by TLN-1 is therefore a possible mechanism to regulate myosin phosphorylation and localization, thereby promoting axon regeneration.

## Methods

### *C. elegans* strains

The *C. elegans* strains used in this study are listed in S2 Table. All strains were maintained on nematode growth medium plates and fed with bacteria of the OP50 strain according to a previously described standard method [38].

### Plasmids

*Punc-25::let-502ΔC* and *Punc-25::venus::mlc-4(DD)* plasmids were previously described [20]. *Punc-25::rtkn-1* and *Pmec-7::rtkn-1* plasmids were generated by inserting the *rtkn-1* cDNA isolated from a cDNA library into the pSC325 and pPD52.102 vectors, respectively. The pDBD-RTKN-1 plasmid was generated by inserting the *rtkn-1* cDNA into the pGBDU vector. pAD-SORB-1 and pAD-SORB-1C plasmids were generated by inserting the *sorb-1* and *sorb-1c* into the pACTII vector. pDBD-ALP-1b(ΔPDZ), pDBD-ALP-1b(ΔLIM4), pDBD-ALP-1b (ΔLIM3) and pDBD-ALP-1b(ΔLIM2) were generated by oligonucleotide-directed PCR using pDBD-ALP-1b as a template and were verified by DNA sequencing. *Punc-25::rho-1(G14V)*, pDBD-ALP-1b, pAD-LET-502ΔC, pAD-MLC-4 and *Pmyo-2::dsred-monomer* plasmids were previously described [20,24].

### Generation of mutants using the CRISPR–Cas9 system

The *tln-1(A2534T)*, *tln-1(e259; T2534A)*, *rtkn-1(km94)*, and *alp-1(km95)* alleles were generated using the CRISPR–Cas9 system as previously described [45]. CRISPR guide RNAs [5′-AACG AGAACUCGGCGAGGCUGUUUUAGAGCUAUGCU-3′ for *tln-1(A2534T)*, 5′-

AACGAGAACUCGGCGAGACUGUUUUAGAGCUAUGCU-3′ for *tln-1(e259; T2534A)*, 5′-GUUGAACAAUAUUCUACCGUGUUUUAGAGCUAUGCU-3′ for *rtkn-1(km94)*, and 5′-AUCCGAACGAGCCAUUCGAAGUUUUAGAGCUAUGCU-3′ for *alp-1(km95)*] and 70 nt single-stranded donor template DNAs [5′-AGGAAGTTCTGCGAAAAGAACGAGAAC TCGGCGAGaCTCGACACAAGCTGGCTCATCTCAATAAGGCTCG-3′ for *tln-1(A2534T)*, 5′-AGGAAGTTCTGCGAAAAGAACGAGAACTCGGCGAGGCTCGACACAAGCTGGCT CATCTCAATAAGGCTCG-3′ for *tln-1(e259; T2534A)*] were synthesized (Integrated DNA Technologies: IDT), and co-injected with the trans-activating CRISPR RNA (IDT), Strepto-coccus pyogenes Cas9 3NLS (IDT) protein, and pRF4(rol-6d) plasmid into KU501 [for *tln-1 (A2534T)*, *rtkn-1(km94)*, and *alp-1(km95)*] and KU1358 [for *tln-1(e259; T2534A)*] strains. Each F1 animal carrying the transgene was transferred to a new dish and single-worm PCR was performed, followed by DNA sequencing to detect mutations.

## Transgenic animals

Transgenic animals were generated using the standard *C. elegans* microinjection method [46]. *Pmyo-2::dsred-monomer*, *Prtkn-1::rtkn-1*, *Punc-25::rtkn-1*, *Pmec-7::rtkn-1*, and *Punc-25::rho-1 (G14V)* plasmids were used in *kmEx1638* [*Prtkn-1::rtkn-1* (25 ng/μL) + *Pmyo-2::dsred-mono-mer* (5 ng/μL)], *kmEx1639* [*Punc-25::rtkn-1* (25 ng/μL) + *Pmyo-2::dsred-monomer* (5 ng/μL)], *kmEx1640* [*Pmec-7::rtkn-1* (25 ng/μL) + *Pmyo-2::dsred-monomer* (5 ng/μL)], and *kmEx1641* [*Punc-25::rho-1(G14V)* (25 ng/μL) + *Pmyo-2::dsred-monomer* (5 ng/μL)], respectively. The *kmEx1405* and *kmEx1406* extrachromosomal arrays were previously described [20]. The *juIs76* array has also been previously described [47].

## Microscopy

Fluorescence images of transgenic animals were observed under a 100x objective lens of a Nikon ECLIPSE E800 fluorescence microscope and photographed with a Zyla CCD camera. Confocal fluorescence images were captured using a Zeiss LSM-800 confocal laser-scanning microscope with a 63x objective.

## Axotomy

Axotomies were performed as previously described [24]. Animals were axotomized at the young adult stage. Commissures with growth cones or small branches present on the proximal fragment were counted as "regenerated". Proximal fragments showing no change after 24 h were counted as "no regeneration". A minimum of 20 individuals with 1–3 axotomized com-missures were observed for most experiments.

## Yeast two-hybrid assays

Yeast two-hybrid analysis was performed as previously described [48]. Plasmids were trans-formed into the *Saccharomyces cerevisiae* reporter strain PJ69-4A (*MAT**a** trp1-901 ura3-52 leu2-3,112 his3-200 gal4Δ gal80Δ Met2::GAL7-lacZ LYS2::GAL1-HIS3 Ade2::GAL2-ADE2*) and the yeasts were grown on SC-Ura-Leu plates. Transformants grown on these plates were then plated out onto SC-Ura-Leu-His plates containing 5 or 10 mM 5-aminotriazole and incubated at 30°C for 4 days.

## Statistical analysis

Statistical analysis was performed as previously described [24]. Briefly, confidence intervals (95%) were calculated using the modified Wald method and two-tailed *P*-values were

calculated using Fisher's exact test (https://www.graphpad.com/quickcalcs/contingency1/) and adjusted by Bonferroni correction for multiple tests.

## Supporting information

**S1 Fig. Structure of the *rtkn-1* mutation.** The nucleotides and corresponding amino acids around the deleted regions are shown. The *rtkn-1(km94)* mutation is a 2-bp deletion that causes a frameshift (amino acids in red) and a premature stop codon (*).
(PDF)

**S2 Fig. Structure of the *sorb-1* mutation.** The nucleotides and corresponding amino acids around the deleted region are shown. The inserted nucleotides are shown in orange. The *sorb-1(gk304)* mutation is a 430-bp deletion and a 2-bp insertion, resulting in a frameshift (amino acids in red) and a premature stop codon (*).
(PDF)

**S3 Fig. Domain structure of DEB-1.** The domain structure of DEB-1 and human vinculin is shown. The *deb-1(gk329549)* allele harbors the *D908V* mutation in the tail domain. Identical and similar residues are highlighted in red and yellow shading, respectively.
(PDF)

**S4 Fig. Structure of the *alp-1* mutation.** The genomic structure of the *alp-1* gene is shown. The nucleotides and corresponding amino acids around the deleted regions are also shown. Nucleotides derived from exonic and intronic regions are indicated by upper- and lower-case letters, respectively. The inserted nucleotides are shown in orange. The *alp-1(ok820)* mutation is a 1,236-bp deletion, resulting in a splicing change that skips an exon and produces a short ALP-1 protein missing 57 amino acids. The *alp-1(km95)* mutation is a 21-bp deletion and a 3-bp insertion, resulting in a frameshift (amino acid in red) and a premature stop codon (*).
(PDF)

**S1 Table. Raw data for genotypes tested by axotomy.**
(PDF)

**S2 Table. Strains used in this study.**
(PDF)

## Acknowledgments

We thank the *Caenorhabditis* Genetic Center (CGC), the National Bio-Resource Project, and the *C. elegans* Knockout Consortium for materials. Some strains were provided by the CGC, which is funded by the NIH Office of Research Infrastructure Programs (P40 OD10440).

## Author Contributions

**Conceptualization:** Yoshiki Sakai, Naoki Hisamoto, Kunihiro Matsumoto.

**Formal analysis:** Yoshiki Sakai, Tatsuhiro Shimizu, Mayuka Tsunekawa, Naoki Hisamoto.

**Funding acquisition:** Naoki Hisamoto, Kunihiro Matsumoto.

**Investigation:** Yoshiki Sakai, Tatsuhiro Shimizu, Naoki Hisamoto, Kunihiro Matsumoto.

**Methodology:** Yoshiki Sakai, Tatsuhiro Shimizu, Naoki Hisamoto.

**Project administration:** Naoki Hisamoto, Kunihiro Matsumoto.

**Resources:** Naoki Hisamoto, Kunihiro Matsumoto.

**Supervision:** Naoki Hisamoto, Kunihiro Matsumoto.

**Validation:** Yoshiki Sakai, Tatsuhiro Shimizu, Naoki Hisamoto.

**Visualization:** Yoshiki Sakai, Naoki Hisamoto, Kunihiro Matsumoto.

**Writing – original draft:** Yoshiki Sakai, Naoki Hisamoto, Kunihiro Matsumoto.

**Writing – review & editing:** Yoshiki Sakai, Naoki Hisamoto, Kunihiro Matsumoto.

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
