## [Decision Letter · Decision Letter 0]

22 Nov 2023

Dear Dr %Matsumoto%,

Thank you very much for submitting your Research Article entitled 'Rhotekin regulates axon regeneration through the talin–vinculin–vinexin axis in  Caenorhabditis elegans' to PLOS Genetics.

The manuscript was fully evaluated at the editorial level and by an independent peer reviewer.  Since we could not obtain additional reviews for the manuscript, despite multiple efforts, in a timely manner, we have opted to move forward with our decision based on the comments of the independent reviewer and editorial review.  Overall, we find the manuscript to provide an advance and based on well executed experiments.  We agree with the concerns that the manuscript's reviewer has articulated in their comments.   We therefore ask you to modify the manuscript according to the review recommendations. Your revisions should address the specific points made by each reviewer.

Yours sincerely,

Kaveh Ashrafi

Academic Editor

PLOS Genetics

Gregory P. Copenhaver

Editor-in-Chief

PLOS Genetics

Reviewer's Responses to Questions

**Comments to the Authors:**

Reviewer #1: This manuscript by Sakai et al. uses C. elegans to integrate the roles of distinct cytoskeletal pathways in axon regeneration. They show that RTKN-1/Rhotekin works independently of Rho and LET-502/ROCK MLC signaling to mediate regeneration, and works with DEB-1/vinculin and SORB-1/vinexin. Importantly, they how that TLN-1/talin might work with both pathways, and that ALP-1/enigma might integrate the activities of the two pathways on the actin cytoskeleton. This leads to a broader understanding of how these distinct pathways work together and are integrated into axon regeneration responses. A careful combination of genetics, genome editing, transgenics, and yeast two hybrid is utilized. Domain specific mutations are strong in. that they point to activity of genes in distinct pathways. For example, showing that activated Rho and MLC-4 cannot rescue RTKN-1 mutations is convincing. The use of genome editing to confirm conclusions is strong. Overall, the data presented are presented well and are backed by robust statistics. The work is an advance in that it integrates known pathways in axon regeneration and should be of broad interest.

Comments:

1) The authors use yeast two hybrid to show that it is the SH3 domain of SORB-1 likely interacts with the PxxP motif of RTKN-1. But this direct interaction is not shown. While it is likely given what is known about these domains, the authors could suggest alternatives in the absence of showing the interaction.

2) The discussion of ALP-1 and TLN-1 (last two results sections) could be explained in a more structured manner. These are complex experiments and interpretations. It might help to break the individual into discrete sections, each with its own heading and conclusion. As presented, it is difficult to follow.

**Have all data underlying the figures and results presented in the manuscript been provided?**

Reviewer #1: Yes

PLOS authors have the option to publish the peer review history of their article (what does this mean?). If published, this will include your full peer review and any attached files.

Reviewer #1: No

---

## [Editor Report · Decision Letter 1]

4 Dec 2023

Dear Dr %Matsumoto%,

We are pleased to inform you that your manuscript entitled "Rhotekin regulates axon regeneration through the talin–vinculin–vinexin axis in  Caenorhabditis elegans" has been editorially accepted for publication in PLOS Genetics. Congratulations!

Yours sincerely,

Kaveh Ashrafi

Academic Editor

PLOS Genetics

Gregory P. Copenhaver

Editor-in-Chief

PLOS Genetics

Comments from the reviewers (if applicable):

**Data Deposition**

http://datadryad.org/submit?journalID=pgenetics&manu=PGENETICS-D-23-01076R1

**Press Queries**

---

## [Editor Report · Acceptance letter]

8 Dec 2023

PGENETICS-D-23-01076R1 

Rhotekin regulates axon regeneration through the talin–vinculin–vinexin axis in  *Caenorhabditis elegans*

Dear Dr Matsumoto, 

We are pleased to inform you that your manuscript entitled "Rhotekin regulates axon regeneration through the talin–vinculin–vinexin axis in  *Caenorhabditis elegans* " has been formally accepted for publication in PLOS Genetics! Your manuscript is now with our production department and you will be notified of the publication date in due course.

With kind regards,

Zsofia Freund

PLOS Genetics

On behalf of:
